

**Transport of Asian trace gases via eddy shedding from the Asian summer monsoon**
**anticyclone and associated impacts on ozone heating rates**
Suvarna Fadnavis[1], Chaitri Roy[1], Rajib Chattopadhyay[1], Christopher E. Sioris[2], Alexandru
Rap[3], Rolf Müller[4], K. Ravi Kumar[5] and Raghavan Krishnan[1]
[1]Indian Institute of Tropical Meteorology, Pune, India
[2]Environment and Climate Change, Toronto, Canada
[3]School of Earth and Environment, University of Leeds, Leeds, United Kingdom
[4]Forschungszentrum Jülich GmbH, IEK-7, Jülich, Germany
[5]King Abdullah University of Science and Technology, Thuwal, Saudi Arabia
*Email of corresponding author: suvarna@tropmet.res.in
**Abstract:**
The highly vibrant Asian Summer Monsoon (ASM) anticyclone plays an important role in
efficient transport of Asian tropospheric air masses to the extratropical upper troposphere and
lower stratosphere (UTLS). In this paper, we demonstrate long-range transport of Asian trace
gases via eddy shedding events using MIPAS (Michelson Interferometer for Passive
Atmospheric Sounding) satellite observations, ERA-Interim re-analysis data and the
ECHAM5–HAMMOZ global chemistry–climate model. Model simulations and observations
consistently show that the Asian boundary layer trace gases are lifted to UTLS altitudes in the
monsoon anticyclone and are further transported horizontally eastward and westward by
eddies detached from the anticyclone. We present an event of eddy shedding during 1-8 July
2003 and discuss a 1995-2016 climatology of eddy shedding events. Our analysis indicates



that eddies detached from the anticyclone are instrumental in distributing the Asian trace gases
away from the Asian region to the West-Pacific (20°-30° N; 120°-150° E) and West-Africa
(20°-30° N, 0°-30° E). Over the last two decades, the estimated frequency of eddy shedding is
~68 % towards West-Africa and ~25 % towards the West-Pacific.
Model sensitivity experiments for a 10 % reduction in Asian emissions of non-methane
volatile organic compounds (NMVOCs) and nitrogen oxides ($NO_x$) were performed with
ECHAM5–HAMMOZ to understand the impact of Asian emissions on the UTLS. The model
simulations show that transport of Asian emissions due to eddy shedding significantly affects
the chemical composition of the upper troposphere (~100-400 hPa) and lower stratosphere
(~100-80 hPa) over West-Africa and the West-Pacific. The 10 % reduction of NMVOCs and
$NO_x$ Asian emissions leads to decreases in peroxyacetyl nitrate (PAN) (2-10 % near 200-80
hPa), ozone (1-4.5 % near ~150 hPa) and ozone heating rates (0.001-0.004 $K \cdot day^{-1}$ near 300-
150 hPa) in the upper troposphere over West-Africa and the West-Pacific.
Key Words: Asian summer monsoon anticyclone; Eddy shedding from the monsoon
anticyclone, Transport of Asian trace gases, Ozone heating rates; ECHAM5-HAMMOZ
model.






## 1. Introduction

Rapid industrialization, traffic growth, and urbanization resulted in significant increases in the concentrations of tropospheric trace gases, such as carbon dioxide ($CO_2$), carbon monoxide (CO) and methane ($CH_4$) over Asia. There is global concern about rising levels of these trace gases (due to their global warming potential) as they are projected to increase further over the coming years despite efforts to implement several mitigation strategies (Ohara et al., 2007). In situ observations, satellite measurements, trajectory analysis and model simulations show long range transport of Asian trace gases to remote locations (e.g. North America, Europe) (Liang et al., 2004). The transported trace gases change the radiative balance, dynamics and chemical composition at the respective locations (Vogel et al., 2016). Satellite observations show increasing trends in several tropospheric Asian trace gases over the last decade, e.g. ozone at ~1-3 % year$^{-1}$ (Verstraeten et al., 2015), CO at 3% year$^{-1}$ (Strode and Pawson, 2013), $NO_x$ at ~3.8 -7.3 % year$^{-1}$ (Schneider and van der A, 2012; Ghude et al., 2013). Biomass burning is another major contributor to the observed growth in these trace gases (van der Werf et al., 2006). Peroxyacetyl nitrate (PAN), a powerful pollutant formed in biomass burning plumes (Wayne, 2000), is a secondary pollutant produced through the oxidation of hydrocarbons released from anthropogenic and biogenic sources. It is a reservoir of reactive nitrogen and plays a fundamental role in the global ozone budget (Tereszchuk et al., 2013; Payne et al., 2017). PAN can also be formed in the upper troposphere through the production of $NO_x$ from lightning (Zhao et al., 2009). Simulations of the Model of Ozone and Related Tracers (MOZART)  show an increase of 20-30 % of PAN concentrations in the upper troposphere and lower stratosphere (UTLS) over the Asian summer monsoon (ASM) region produced from



lightning (Tie et al., 2002). While in the lower troposphere, PAN has a short lifetime (a few
hours), in the UTLS it has a longer lifetime (3-5 months), and can therefore act as a reservoir
and carrier of $NO_x$ (Tereszchuk et al., 2013). Recent satellite observations show an increasing
trend in PAN (~0.1 ± 0.05 to 2.7 ± 0.8 ppt year$^{-1}$) in the UTLS over Asia (Fadnavis et al.,

68   2014).

Monsoon convection plays an important role in lofting of boundary layer Asian air masses to
the UTLS (e.g., Randel et al., 2010; Fadnavis et al., 2015; Santee et al., 2017). The uplifted air
masses become confined into the anticyclone enclosed by jets (westerly and easterly jets to the
north and south, respectively), which act as a strong transport-barrier and restrict isentropic
mixing into the extra-tropical lower stratosphere or the equatorial tropics (Ploeger et al., 2015;
Ploeger et al., 2017). Confinements of high amounts of trace gases, including ozone precursors
(e.g., hydrogen cyanide (HCN), CO, hydrochloric acid (HCl), $NO_x$ and PAN), and low ozone
in the anticyclone are evident in satellite and aircraft observations, (Randel et al., 2010; Vogel
et al., 2014; Fadnavis et al., 2015; Ungermann et al., 2016; Santee et al., 2017). The observed
ozone minimum in spite of high amounts of its precursors in the anticyclone is still an open
question. The trace gases partially enter the lower stratosphere and affect the UTLS chemical
composition (Randel et al., 2010; Fadnavis et al., 2015, 2016; Garny and Randel, 2016), with
associated radiative forcing impacts (Riese et al., 2012). Cross-tropopause transport associated
with the Asian monsoon is evident in a number of species, including aerosols, hydrogen
cyanide (HCN) and PAN (Randel et al. 2010; Fadnavis et al. 2014, 2015; Bourassa et al.,

84   2012).



The ASM anticyclone is highly dynamic in nature (e.g., Hsu and Plumb, 2000; Popovic and
Plumb, 2001; Vogel et al., 2016). On the sub-seasonal scale, it shows variation in strength and
location (Garny and Randel, 2016). It frequently sheds eddies and on occasions, it splits into
two anticyclones, namely the Tibetan and Iranian anticyclones (Zhang et al., 2002; Nützel et
al., 2016). An eddy detached from the anticyclone carries Asian air masses (trace gases) away
from the ASM region. There are scattered studies indicating eddy shedding to the west
(Popovic and Plumb, 2001) and east (Ungermann et al., 2016; Vogel et al., 2014) of the
anticyclone. An eddy shedding event causes irreversible mixing in the surrounding air
changing the chemical composition and radiative balance of that region (Garny and Randel,
2016). Here, we analyze in detail transport of Asian trace gases via eddies, subsequent mixing
into the extra-tropics and radiative impact of eddy shedding events on decadal scales. In this
paper, we ask the following questions: (1) how frequent were eddy shedding events during the
last two decades? (2) Which regions are the most affected? (3) Does the transport of Asian
trace gases arising from eddy shedding affect UTLS ozone concentrations and heating rates  at
remote locations?
To address these questions, we first consider an eddy shedding event demonstrating eastward
and westward shedding from the ASM anticyclone during 1-8 July 2003. This year was chosen
since the monsoon season was quite normal (i.e., no evidence of El Niño or Indian Ocean
dipole phenomenon influencing the monsoon circulation). We then present a climatology of
eddy shedding events and lead-lag relations of eddies with the anticyclone. We also evaluate
the impact of increasing Asian emissions of $NO_x$ and NMVOCs on ozone and PAN during the
eddy shedding event, using model sensitivity simulations. Finally, we estimate the associated



changes in ozone heating rates in the UTLS due to Asian trace gases transported via eddy
shedding events.
**2. Experimental set-up and satellite observations**
**2.1 Satellite observations**
The MIPAS (Michelson Interferometer for Passive Atmospheric Sounding) instrument was
launched in March 2002 into a polar orbit of 800 km altitude.  Its orbital period is about 100
min. MIPAS-E provided continual limb emission measurements in the mid-infrared over the
range 685– 2410 cm$^{-1}$ (14.6–4.15 μm) until December 2004 (Fischer et al., 2008). From
January 2005 through April 2012, MIPAS-E was operated with a reduced spectral resolution,
and a vertical resolution of 3 km of in the UTLS region. MIPAS monitored many atmospheric
trace constituents including CO, PAN, and O$_3$. The details of the general retrieval method and
setup, error estimates and use of averaging kernel and visibility flag are documented by von
Clarmann et al. (2009). Here, we analyze the MIPAS observed CO, PAN, and O$_3$ data during
1-8 July 2003.
To account for the comparatively low, and altitude-dependent vertical resolution of MIPAS,
the model data were convolved with the MIPAS averaging kernel to be directly comparable to
MIPAS measurements of CO, PAN, and ozone. MIPAS vertical resolution for CO, O$_3$ and
PAN in the UTLS is 5, 3.5 and 5 km, respectively. The data are contoured and gridded at 15°
latitude and 10° longitude resolution. In the process, the data quality specifications as
documented     at     http://share.lsdf.kit.edu/imk/asf/sat/mipas-export/Documentation/     were



employed, namely: only data with a visibility flag equal to 1 and a diagonal value of averaging
kernel greater than 0.03 were used for ozone and PAN, while 0.008 was used for CO.
**2.2 Experimental set-up**
We employ the ECHAM5-HAMMOZ (Roeckner et al., 2003) aerosol-chemistry-climate
model to understand re-distribution of Asian trace gases via eddy shedding from the
anticyclone. ECHAM5-HAMMOZ comprises of the general circulation model ECHAM5
(Roeckner et al., 2003), the tropospheric chemistry module MOZ (Horowitz et al., 2003), and
the aerosol module, Hamburg Aerosol Model (HAM) (Stier et al., 2005). The chemistry of
ozone, VOCs, $NO_x$, and other gas-phase species is based on the MOZART-2 chemical scheme
(Horowitz et al., 2003). It includes $O_x$-$NO_x$-hydrocarbons with 63 tracers and168 reactions.
The details of the parameterizations and emissions used in the model as well as a validation of
the results are described by Fadnavis et al. (2013, 2014, 2015) and Pozzoli et al. (2011).
The model simulations were performed at the T42 spectral resolution corresponding to about
$2.8° \times 2.8°$ in the horizontal dimension and 31 vertical hybrid σ-p levels from the surface up to
10 hPa. Here, we note that our base year for aerosol and trace gas emissions is 2000. We
performed two simulations: (i) a control experiment (CTRL), and (ii) a sensitivity experiment
(Asia10), where emissions of both $NO_x$ and NMVOCs were simultaneously reduced by 10 %
over Asia (10° S–50° N, 60–130° E). Both simulations were performed for the year 2003
driven by European Centre for Medium-Range Weather Forecasts operational analyses
(Integrated Forecast System (IFS) cycle-32r2) meteorological fields (available every six
hours) (Uppala et al., 2005). All simulations include lightning $NO_x$ and the subsequent PAN





production. Since the lightning parameterization is the same in the CTRL and sensitivity
simulations, its impact may be negligible. However, there may be an indirect impact of
changed emissions on lightning and thus on $NO_x$ or PAN production. The model simulations
used here are the same as those used by Fadnavis et al. (2015).
The climatology of ozone mass mixing ratio, winds and Potential Vorticity (PV) are obtained
from ERA-Interim reanalysis data for the period 1995-2016. The anomalies are obtained from
difference between daily mean values of July 2003 and daily climatology. Power spectral
analysis and lag/lead correlations have been carried out on PV data for the period 1995-2016
to show climatological features.
Instantaneous ozone heating rates are calculated using the Edwards and Slingo (1996)
radiative transfer model. We used the off-line version of the model, with six shortwave and
nine longwave bands, and a delta-Eddington 2-stream scattering solver at all wavelengths, in a
set-up similar to other recent studies (Rap et al., 2015, Roy et al., 2017).
**3. Results**
**3.1 A typical case study of eddy shedding from the monsoon anticyclone**
The dynamics of the monsoon anticyclone is better portrayed at the 370 K potential
temperature surface and the monsoon anticyclone is obvious as an area of low PV values (PV<
2 PVU, 1 PVU = $10^{-6}$ K $m^2$ $kg^{-1}$ $s^{-1}$) (indicating tropospheric air-mass) at this surface (Garny
and Randel, 2016). Eddies are identified as air with low PV emanating from the monsoon
anticyclone (Popovic and Plumb, 2001; Vogel et al., 2014). Past studies have shown that





during the monsoon season (June to September), the bulk of the low PV air at the isentropic
level of 370 K, is confined between about 20–35° N and 20–120° E indicating the spatial
extent of the anticyclone (Popovic and Plumb, 2001; Vogel et al., 2014; Garny and Randel,
2016). A pocket of low PV air-mass detached from the boundary of the anticyclone (outside
the anticyclone, 20–35° N and 20–120° E) is considered as an eddy. **Figure 1a-h** shows the
distribution of PV at 370 K during 1-8 July 2003. It can be seen that during this period the
anticyclone was wobbling and shed eddies eastward and westward over West-Africa (20-30°
N, 0-30° E) and the West-Pacific (20-30° N; 120-150° E).  Initially, during 2-5 July 2003, the
ASM anticyclone shed an eddy westward over West-Africa. The eddy moved further west
with the progression of time. Later during 4-8 July 2003, eddy shedding occurred to the east of
the anticyclone, over the West-Pacific and the air detached from the anticyclone moved further
eastward with time. The longitude-pressure section of PV shows that the eddy protrudes down
to 400 hPa (not shown).
Previous studies have shown that eddy shedding events are associated with Rossby wave
breaking (RWB) (Hsu and Plumb, 2000; Popovic and Plumb, 2001; Fadnavis and
Chattopadhyay, 2017). The RWB is manifested as a rapid and large-scale irreversible
overturning of PV contours on the 350K isentropic surface. It is accompanied with a cyclonic
circulation at 200 hPa (Strong and Magnusdottir, 2008; Fadnavis and Chattopadhyay, 2017).
**Figure 2a-h** shows the distribution of PV at the 350K surface and the circulation at 200 hPa
during 1-8 July 2003. It can be seen that, during 1-8 July 2003, three RWB events occurred:
one near 30° E (referred as RWB-1), one near 70° E (refereed as RWB-2) and another one





near 120°E (referred to as RWB-3). Since RWB-3 was outside the region of the ASM

anticyclone (over the West-Pacific ∼150-170° E) it did not play a role in the eddy shedding

event of 1-8 July. If we track the location of these RWB events (indicated by the black and red

arrows), one can see that, with the progression of time, the RWB feature moved eastward. The

eastward migration of RWB is linked to its movement along the subtropical westerly jet

(Fadnavis and Chattopadhyay, 2017). Initially during 1-5 July RWB-1 was strong (PV > 2

PVU) while RWB-2 (PV < 2 PVU) was weak. During this period the southward and westward

moving RWB-1 leads to eddy shedding over West Africa. Later, during 4-8 July, RWB-2

strengthened while RWB-1 weakened and disappeared. The southward and eastward moving

RWB-2 was responsible for the eddy shedding event near the Western Pacific (see **Fig. 2d-h**).

**3.2.    Climatology of eddy shedding from the monsoon anticyclone**

A power spectrum analysis has been performed on the PV data (averaged for 300-100 hPa)

during 1995-2016 for West-Africa (20-30° N, 0-30° E) and the West-Pacific (20-30° N, 140-

150° E). **Figure 3a-b** shows the distribution of power spectral variance over these two regions.

The variance is significant at 99 % for 3-5 days and 12-15 days for both the regions indicating

that the eddy shedding activity is dominated in the range of synoptic frequency (∼10 days).

Popovic and Plumb (2001) also indicated a typical duration of an eddy shedding event of ∼4-8

days. We compute the frequency of eddy shedding days (PV < 1 PVU) occurring over West-

Africa and the Western Pacific. The ERA-Interim data for the last two decades show that eddy

shedding is quite frequent over west-Africa ∼68 % and the West-Pacific ∼25 %.  The lag-lead

correlation of PV (averaged for 200-100 hPa) for the centre region of the anticyclone (85-90°

E, 28-30° N) with PV averaged over the West-Pacific shows a maximum positive lead
correlation at 3-4 days (**Fig. 3c**). Similarly, PV over West-Africa shows a maximum positive
lead correlation for 5-6 days with the PV averaged over the monsoon anticyclone (**Fig. 3d**).
This indicates that the transport of the eddies from the anticyclone (source region) has a
typical duration of three to four days over the West Pacific and five to six days over West
Africa. This transport time is the timescale over which the trace gases are moved to remote
locations from the ASM anticyclone.
**3.3. Long range transport of trace gases**
**3.3.1   Horizontal transport of ozone, CO and PAN via eddies**
Biomass burning over south-east Asia and East Asia produces large amounts of CO, $NO_x$,
VOCs, PAN, ozone and aerosols (e.g., Streets et al., 2003, Fadnavis et al., 2014). The
monsoon convection over the Bay of Bengal, southern slopes of Himalaya and South China
Sea (see **Fig. S1**) lifts up these species into the anticyclone where they may get dispersed in
the UTLS by the vibrant anticyclone and its associated eddies. **Figure 4a-h** shows the
distribution of ozone during 1-8 July 2003 (MIPAS $O_3$ is binned for 2 days and simulated $O_3$
is plotted for alternate days) in the anticyclone at 16 km (~100 hPa). Ozone concentrations
from MIPAS satellite measurements and model simulations (CTRL) are plotted at 16 km and
from ERA-Interim reanalysis at 100 hPa. For comparison, we have interpolated the model data
to the MIPAS altitude grid and smoothed with the averaging kernel. The ASM anticyclone is
marked by minimum ozone although its precursors (e.g. CO, $NO_x$ and $CH_4$) show maxima
(Randel et al., 2010; Roy et al., 2017). The spatial pattern of low ozone amounts in the



anticyclone and the associated eddies is evident in all of the data sets during 1-8 July 2003.
During 1-5 July, ozone concentrations in the eddy over West-Africa are ~60-200 ppb in
MIPAS, ~60-100 ppb in ERA-Interim and 80-150 ppb in the model simulations. During 4-8
July, the eddy over the west Pacific shows ozone amounts of ~80-200 ppb in MIPAS, ~80-120
ppb in ERA-Interim and ~150-200 ppb in the model simulations. In general, MIPAS
measurements and simulated ozone amounts shows reasonable agreement, while ozone
amounts in ERA-Interim are lower by 30-80 ppb than both MIPAS measurements and the
model simulations.
**Figure 5a-h** shows the distribution of CO from MIPAS observations and model simulations
during 1-8 July 2003 (MIPAS CO is binned for 2 days and simulated CO is plotted for
alternate days). The confinement of high concentrations of CO in the anticyclone and in eddies
is seen in both MIPAS observations and model simulations. During 1-5 July, eddies over west-
Africa and west-Pacific show CO volume mixing ratios of ~65-85 ppb in MIPAS, and ~70-90
ppb in the model simulations.
**Figure 6a-h** shows the distribution of PAN from MIPAS measurements and the model
simulation (CTRL) at 16km during 1-8 July 2003 (MIPAS PAN mixing ratios are binned for
2 days and simulated PAN is plotted for alternate days). A confinement of high amounts of
PAN in the anticyclone and the associated eddies is seen both in the MIPAS measurements
and the model simulations. During 1-5 July, MIPAS observed PAN amounts are ~120-230 ppt
in eddies over west-Africa, while the model simulation shows ~180-240 ppt of PAN at the



same location. The eddy over the west-Pacific shows PAN amounts of ~120-230 ppt in the
MIPAS measurements and 160-230 ppt in the model simulations.
There are minor differences in ozone, CO and PAN amounts from model simulation, satellite
observations and ozone from ERA-Interim. These differences may be due to a number of
reasons e.g. different grid sizes of MIPAS ($10° \times 15°$), ERA-Interim ($0.75° \times 0.75°$) and model
data ($2.8° \times 2.8°$), binning of MIPAS data for two days to accommodate better special
coverage, uncertainties in the model emission inventory, and retrieval errors in the satellite
data.
**3.3.2 Vertical distribution of CO, PAN and ozone**
Further, we show the vertical distribution of CO and PAN as an indication of the Asian
biomass burning emissions. **Figure 7** shows longitude-pressure cross-sections (averaged for
20°-40° N) of CO and PAN from the CTRL simulation, with wind vectors depicting
circulation patterns. It shows during 1-5 July 2003 a plume of CO/PAN uplifted from the
Asian region (80°-120° E) moving further upward into the UTLS. The location of the plume
coincides with the region of convective transport (**Fig. S1**). In the upper troposphere (~120
hPa) westward horizontal transport of CO/PAN towards West-Africa is obvious as a result of
eddy shedding during the respective days. In particular, during 2-4 July high amounts of
CO/PAN are observed near 0°-30° E at 100 hPa (**Fig. 7a-b** and **7e-f**). On 2 July there is some
PAN transport over west-Pacific. During 4-8 July 2003, eddy shedding occurs to the east of
the anticyclone over the West-Pacific (120°-150° E) (see Figure 1e-f). East-ward horizontal
transport of CO/PAN in the regions of eddy shedding is evident in **Fig.7c-d** and **7g-h**. The



Asian trace gases then disperse downward deep into the troposphere (~500 hPa over the West
Pacific and ~200 hPa over West-Africa) and are partially lifted into the lower stratosphere.
The vertical distribution of ozone shows low ozone amounts extending from convective
regions of the Bay of Bengal (80-95° E) and the South China Sea (~120° E) upward in the
upper troposphere (**Fig.S2**). This is due to low ozone amounts in marine air masses over Asia
during the monsoon season (Zhao et al., 2009). This feature is not as clear as seen in the
vertical distribution of CO and PAN, since a number of factors are influencing ozone
production and loss processes at different altitudes in the troposphere and lower stratosphere
(e.g. lightning in the upper troposphere).
**3.4 Influence of Asian emissions on extra-tropical UTLS**

In this section, we investigate the influence of Asian anthropogenic emissions of

NMVOCs and $NO_x$ on the distribution of PAN and ozone in the tropical/extra-tropical UTLS
from sensitivity experiments. **Figure 8a-d** shows anomalies of PAN (Asia10-CTRL) at 16km
during 1-8 July 2003 (plotted on alternate days). The negative anomalies in PAN are seen
confined to the region of the anticyclone and the associated eddies (1-5 July over West-Africa
and 4-8 July over West-Pacific). These anomalies portray the response of Asian boundary
layer emissions (NMVOCs and $NO_x$) on the upper level anticyclone and the associated eddies.
A number of studies (Randel et al., 2010; Fadnavis et al., 2013; 2015; Vogel et al., 2014) have
shown lifting of Asian emissions to the UTLS by the monsoon convection and its confinement
in the anticyclone. Decrease in Asian emissions (NMVOCs and $NO_x$) by 10 % decreases PAN



amounts by ~5-23 % in the ASM anticyclone and the associated eddies over West-Africa and
the West-Pacific.
Further, we analyze the vertical distribution of anomalies of PAN and ozone. Figure 8e-h
shows longitude-pressure sections of anomalies of PAN. It shows negative anomalies (in
response to reduced Asian emissions) along the transport pathways (**Fig. S1**), i.e., from the
boundary layer of the Asian region (80°-120° E) into the upper troposphere and
westward/eastward transport from the anticyclone owing to eddy shedding. These anomalies
extending above the tropopause indicate cross-tropopause transport. Our simulations show that
a 10 % reduction in Asian emissions of both NMVOCs and $NO_x$, results in a decrease in the
amount of PAN by ~2-10 % over North-West Africa during 1-5 July and over the Western
Pacific during 4-8 July 2003.
The vertical distribution of ozone anomalies show negative values (–1 to –4.5 %) in the
troposphere extending from the surface up to 180 hPa along the transport pathways (80°-110°
E). Near the tropopause ozone anomalies are positive, varying between 1 to 8 % (**Fig.8i-l**).
During the monsoon season, marine air masses containing low amounts of ozone prevail over
the Asian land mass. The monsoon air mass gathers Asian boundary layer ozone precursors
(and the other trace gases) and is uplifted to the UTLS by the monsoon circulation. It should
be noted that a decrease in emissions of $NO_x$ and NMVOCs in the Asia10 simulations
produces lower ozone amounts in the troposphere than CTRL. Therefore, in the regions of
eddy shedding, negative anomalies near 200-300 hPa indicate transport of monsoon air (via
eddies) towards West-Africa during 1-5 July and to the West-Pacific during 4-8 July. Also,



there may be ozone production in the troposphere from its precursors carried by the monsoon
circulation. PAN transported by eddies in the troposphere over West-Africa and the West
Pacific will release NO$_x$ and contributes to tropospheric ozone production (**Fig. 7e-h** and **Fig.**
**8e-h**). The increase in lower stratospheric ozone concentrations near the tropopause (Fig. 8i-l),
in response to the 10 % reduction of Asian NO$_x$ and NMVOCs emissions, may be due to the
inverse relation between ozone amounts and its precursors in the monsoon anticyclone
(Randel et al., 2010) and other factors such as changes in dynamics due to emission change.
Ozone changes near the tropopause have been shown to have a large impact on the Earth's
radiative balance (Riese et al., 2012).
**3.5 Influence of Asian emission of trace gases on ozone heating rates**
Ozone is a dominant contributor to radiative heating in the tropical lower stratosphere,
impacting the local heating budget and non-local forcing of the troposphere below (Gilford
and Solomon, 2017). We estimate changes in ozone heating rates caused by a 10 % decrease
in Asian NMVOCs and NO$_x$ emissions. **Figure 9a-d**, showing anomalies of ozone heating
rates on 1-8 July (plotted on alternate days), indicates a reduction in ozone heating rates in
response to a decrease in Asian NMVOCs and NO$_x$ emissions, coincident with the region of
convective transport (**Fig. S1**). In the upper troposphere (300-180 hPa), the negative anomalies
in ozone heating rates vary between –0.001 and –0.0045 K·day$^{-1}$. Interestingly, reduced Asian
emissions (NMVOCS and NO$_x$), leads to a reduction in ozone, which leads to a reduction in
ozone heating rates (–0.001 to –0.003 K·day$^{-1}$) in the region of eddy shedding over West-
Africa (1-5 July) and the West-Pacific (4-8 July). The ozone poor Asian air mass trapped



within eddies has reduced the heating over West-Africa and the West-Pacific. Influence of
Asian $NO_x$ emissions on ozone heating rates (mean for June-September ~0.0001 - 0.0012
$K \cdot day^{-1}$ for 38 % increase over India) in the upper troposphere (300-200 hPa) have been
reported in the past (Roy et al., 2017). Near the tropopause, ozone heating rates are positive
0.001 - 0.005 $K \cdot day^{-1}$, which is due to positive anomalies of ozone near the tropopause (**Fig.**
**8i-l**). The ECMWF dataset for 44 years (1958-2001) shows an inter-annual amplitude of the
ozone heating rate ±0.00025 $K \cdot day^{-1}$ near the tropopause over 30° S-30° N (Wang et al. 2008).
**4**. **Summary and Discussion**
In this study we show evidence of eddy shedding from the ASM anticyclone to both its eastern
and western edge, during 1-8 July 2003 based on MIPAS satellite observations and ERA
Interim re-analysis data as well as the associated transport patterns of trace gases from the
ASM region to remote regions. The transport diagnostic based on ERA-Interim data shows
that eddy shedding events are associated with RWB in the subtropical westerly jet. The RWB
feature moves eastward in the subtropical westerly jet. Initially, during 1-5 July 2003, RWB
occurs in the western part of the anticyclone and then sheds over West-Africa (20°-30° N, 0°-
30° E). Later, during 5-8 July 2003, RWB moves to the eastern part of the anticyclone and
sheds an eddy over the West-Pacific (20°-30° N; 120°-150° E). Analysis of ERA-Interim PV
data for the last two decades (1995-2016) shows that the frequency of eddy shedding from the
ASM anticyclone over West-Africa is ~68 % and ~25 % over the West-Pacific. PV (300-100
hPa) at the centre of the anticyclone (85°-90° E, 28°-30° N) shows maximum correlation with
PV over West-Africa 3-4 days later and 5-6 days later in the West-Pacific. It indicates that the



anticyclone sheds eddies with transport duration of typically three to four days to West Africa
and five-six days to the Western Pacific.
We employ the chemistry climate model ECHAM5-HAMMOZ to investigate transport of
Asian boundary layer trace gases (CO, ozone and PAN) into the monsoon anticyclone and the
associated eddies. The model simulations show that Asian trace gases transported into the
monsoon anticyclone are further carried away horizontally towards West-Africa and the West-
Pacific by eddies which detach from the anticyclone. These eddies protrude down to ~200 hPa
over West-Africa and ~500 hPa over the West Pacific. They re-distribute Asian trace gases
downward into the troposphere over these regions. Moreover, part of this air-mass is also
transported upward into the lower stratosphere. A higher frequency of eddy shedding over
West-Africa (68 %) during last two decades indicates a greater influence of Asian trace gases
on the UTLS over West-Africa than the West-Pacific over last two decades (1995-2016).

We evaluate the impact of Asian $NO_x$ and NMVOCs emissions on ozone and PAN in

the regions of the ASM anticyclone and the associated eddies. The model sensitivity
simulations for a 10 % reduction in Asian emissions of NMVOCs and $NO_x$ indicate significant
reduction (~2-10 %) in the concentration of PAN in the UTLS (300-80 hPa) over West-Africa
and the West-Pacific. The vertical distribution of anomalies of PAN shows negative values
along the transport pathways, i.e., rising from the Asian region (80°-120° E) into the upper
troposphere and both westward and eastward transport towards the region of eddy shedding.
Tropospheric ozone (1000-180 hPa) shows a decrease of up to –4.5 % in response to a 10 %
decrease in Asian emissions of NMVOCs and $NO_x$, while positive ozone anomalies (up to 8



%) are seen near the tropopause. The reason for the observed ozone minimum (noting that
ozone precursors show high amounts) in the anticyclone is still an open question. The satellites
and aircraft observations show inverse relation between the amount of ozone and its
precursors. The increase in ozone anomalies in the anticyclone in response to a reduction of
$NO_x$ and NMVOCs may be a consequence of the observed inverse relation between ozone and
its precursors in the anticyclone or it may be due to changes in dynamics in response to
emission change, which requires further investigations.
Our analysis indicates that transport of Asian trace gases from the anticyclone to West-Africa
and the West-Pacific via eddies causes a change in the chemical composition of the UTLS and
may therefore impact the radiative balance of the UTLS. We also estimate that a 10 %
reduction in Asian NMVOCs and $NO_x$ emissions leads to a decrease of ozone heating rates of
0.001 to 0.004 K·day$^{-1}$ in the region of transport into the troposphere and an increase of 0.001
to 0.005 K·day$^{-1}$ near the tropopause and lower stratosphere (180-50 hPa) over Asia (20°-150°
E; 20°-40° N). Previous studies show that ozone changes in the lower stratosphere have the
largest impact on the ozone radiative forcing (Riese et al., 2012). Interestingly, in the upper
troposphere (200-300 hPa) negative anomalies of ozone heating rates (~0.001-0.003 K·day$^{-1}$)
are seen in the region of eddy shedding over West-Africa and the West-Pacific. Thus transport
of Asian air masses via eddies eventually alters the heating rates in the UTLS in the regions of
eddy shedding and may thus affect radiative forcing and local temperature. However such
questions are beyond the scope of this study. It should be noted that there are minor
differences in the amounts of ozone, CO and PAN between model simulations and satellite
observations (ozone from ERA-Interim). The ozone heating rates estimated from the model



simulations will vary accordingly. Notwithstanding, we suggest further scrutiny of long range
transport of Asian trace gases via eddies shedding from the anticyclone and its impact on
ozone heating rates in the respective regions.
**Acknowledgements:** Dr. S. Fadnavis and C. Roy acknowledges with gratitude Prof. Ravi
Nanjundiah, Director of IITM, for his encouragement during the course of this study. We are
grateful to B. Vogel for helpful discussions. This work was partly funded by the European
Community's Seventh Framework Programme (FP7/2007–2013) as part of the StratoClim
project (grant agreement no. 603557). We thank the European Centre for Medium-Range
Weather Forecasts (ECMWF) for providing meteorological data sets. The authors are also
thankful to Dr. Bernd Funke and Dr. Michael Kiefer for their help related to MIPAS data.



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

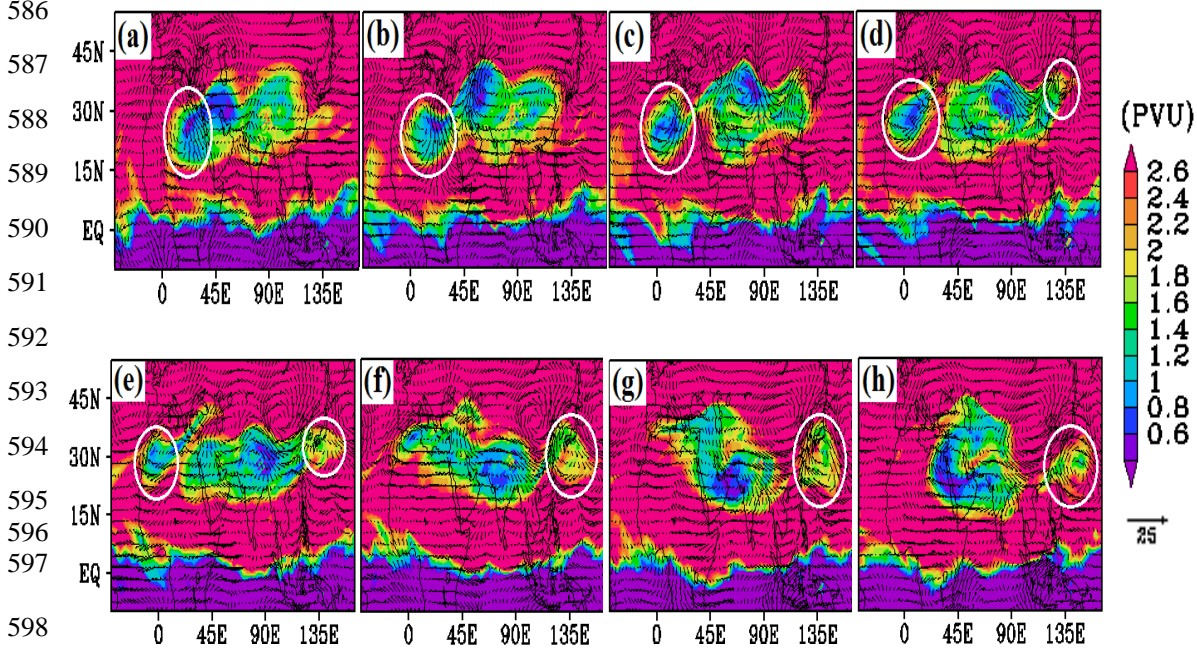

Figure 1: Spatial distribution of potential vorticity (PVU) (1 PVU = $10^{-6}$ K m$^2$ kg$^{-1}$ s$^{-1}$)

(color shades) at 370 K potential temperature surface and wind anomalies at 200 hPa

from ERA-Interim reanalysis for (a) 01 July, (b) 02 July, (c) 03 July, (d) 04 July,  (e) 05 July, (f)

06 July, (g) 07 July, (h) 08 July, 2003. Wind vectors are represented by black arrows (m·s$^{-1}$).

Eddies are shown with white circles.



605

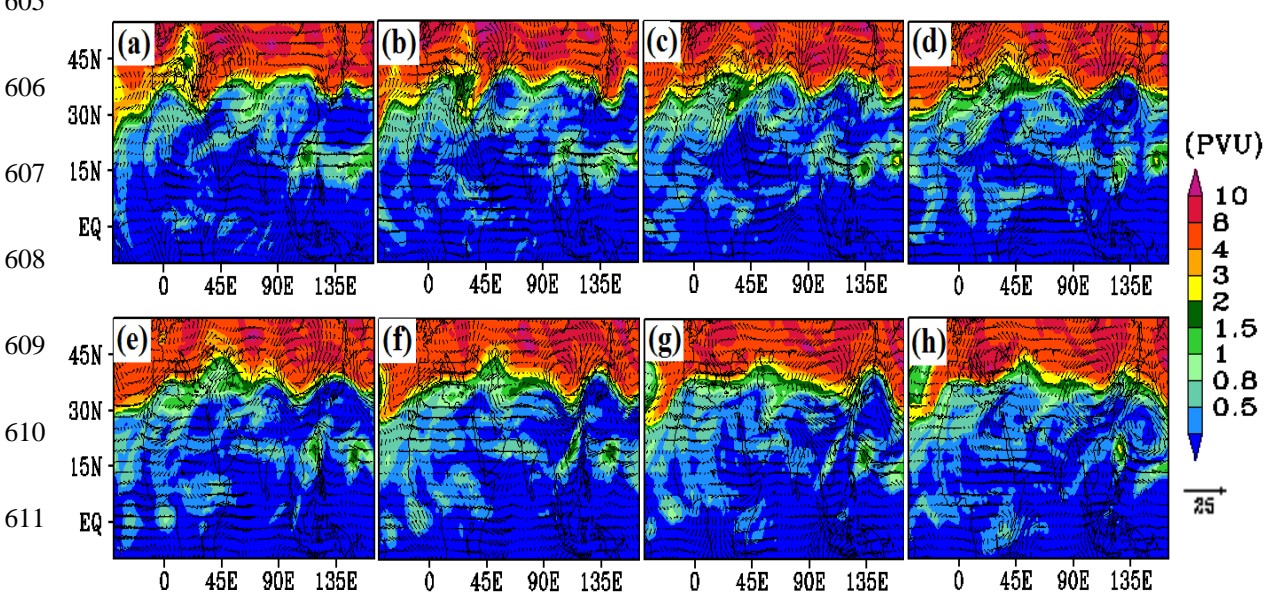


Figure 2: Spatial distribution of potential vorticity (PVU) (color shades) at 350 K level and
wind anomalies in m·s$^{-1}$ (thin black vectors) at 200 hPa from ERA-Interim reanalysis for (a)
01 July, (b) 02 July, (c) 03 July, (d) 04 July,  (e) 05 July, (f) 06 July, (g) 07 July, (h) 08 July,
2003. The events of RWB-1, RBW-2 and RWB-3 are indicated by solid black, red and blue
arrows, respectively.





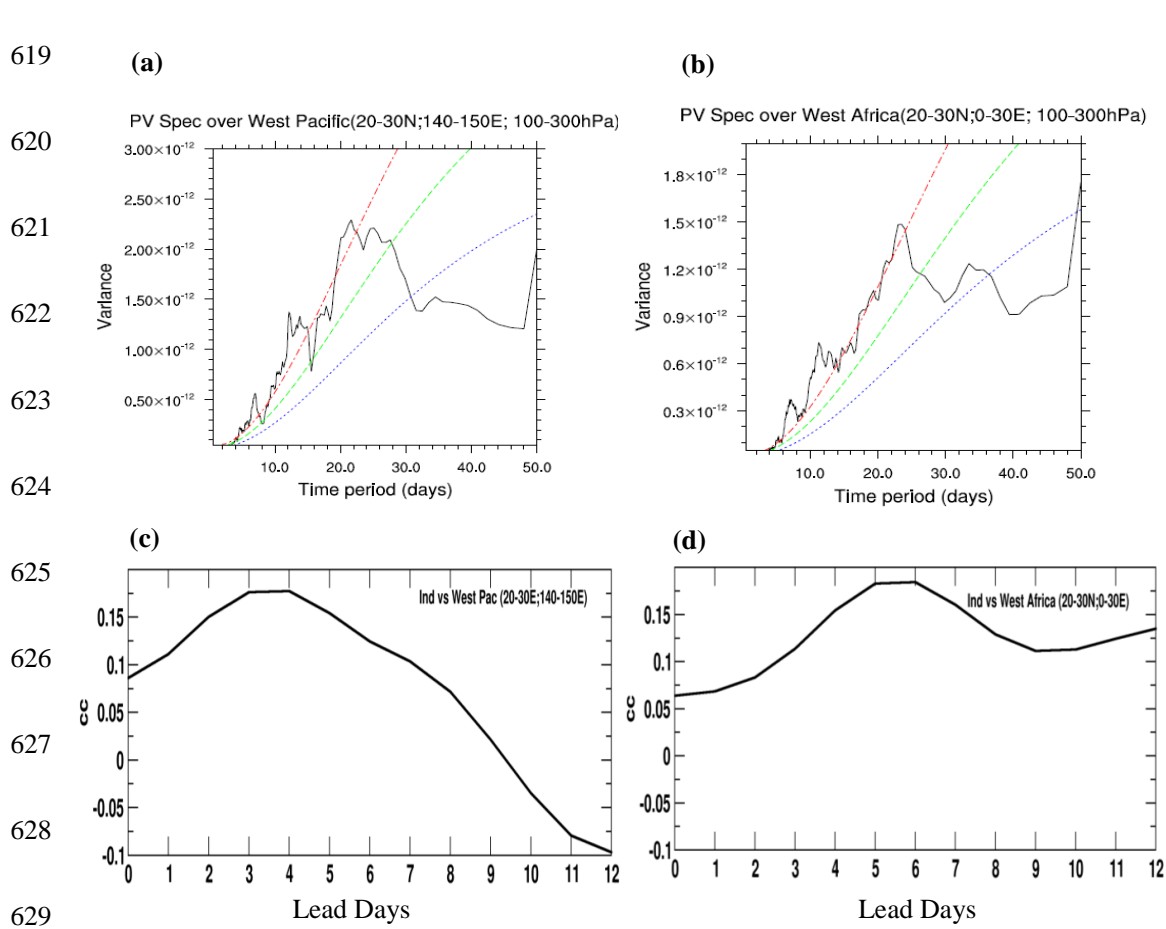

Figure 3: Power spectral analysis of ERA-Interim PV averaged for 100-300 hPa and in June-
September during 1995-2015 (a) West-Africa (20-30° N, 0-30° E) and (b) West-Pacific (20-
30° N, 140-150° E) and lag-lead correlation of PV in the monsoon anticyclone (85-90° E, 28-
30° N) with (c) West-Pacific (20-30° N, 140-150° E), (d) West Africa (20-30° N, 0-30° E).

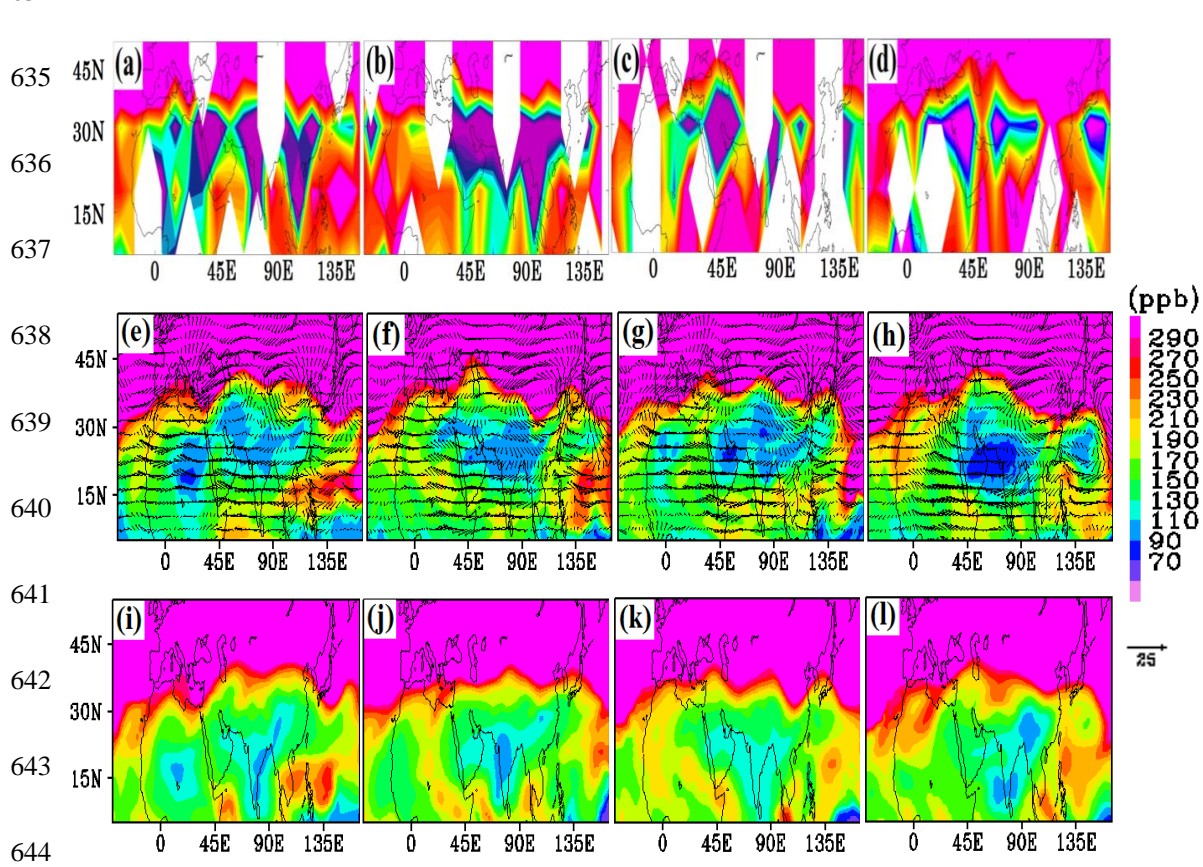

Figure 4: Spatial distribution of ozone mixing ratios (ppb) (color shades) corresponding to
MIPAS satellite observations at 16 km for (a) 1-2 July, (b) 3-4 July, (c) 5-6 July, (d) 7-8 July,
2003;  ERA-Interim reanalysis at 100 hPa for (e) 2 July, (f) 4 July, (g) 6 July, (h) 8 July, 2003,
and ECHAM5-HAMMOZ CTRL simulations at 16 km for (i) 2 July, (j) 4 July, (k) 6 July, (l) 8
July, 2003. Black arrows in panels (e)-(h) show wind anomalies (m·s$^{-1}$) at 200 hPa.



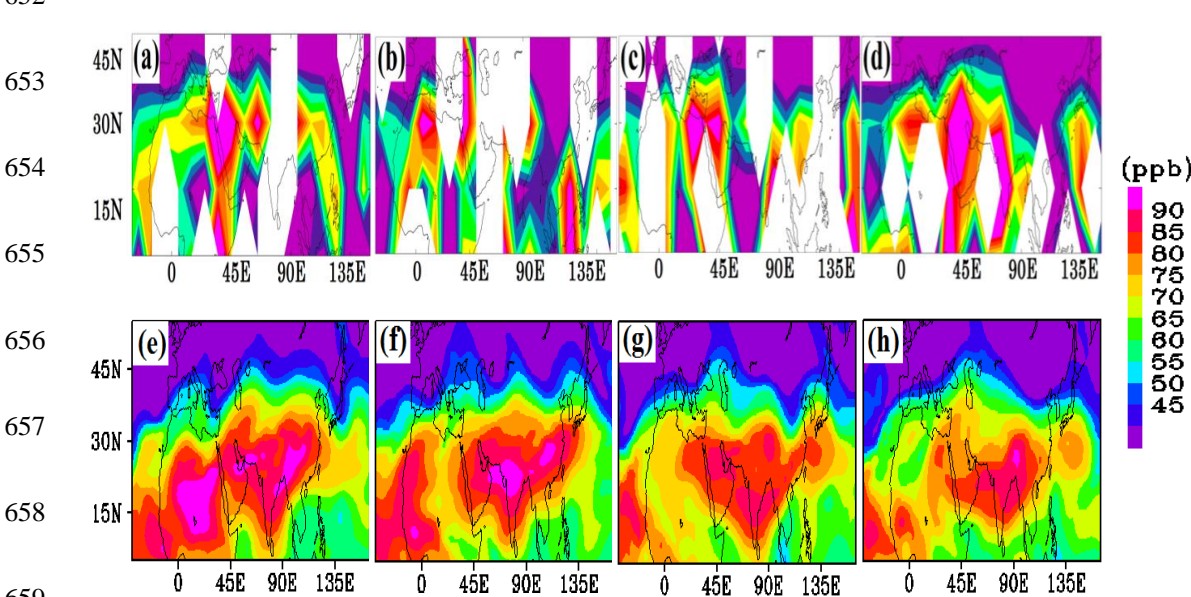

Figure 5: Spatial distribution of CO mixing ratios (ppb) at 16 km: MIPAS satellite
observations for (a) 1-2 July, (b) 3-4 July, (c) 5-6 July, (d) 7-8 July, 2003 and ECHAM5-
HAMMOZ CTRL simulations for (e) 02 July, (f) 04 July, (g) 06 July, (h) 08 July, 2003.






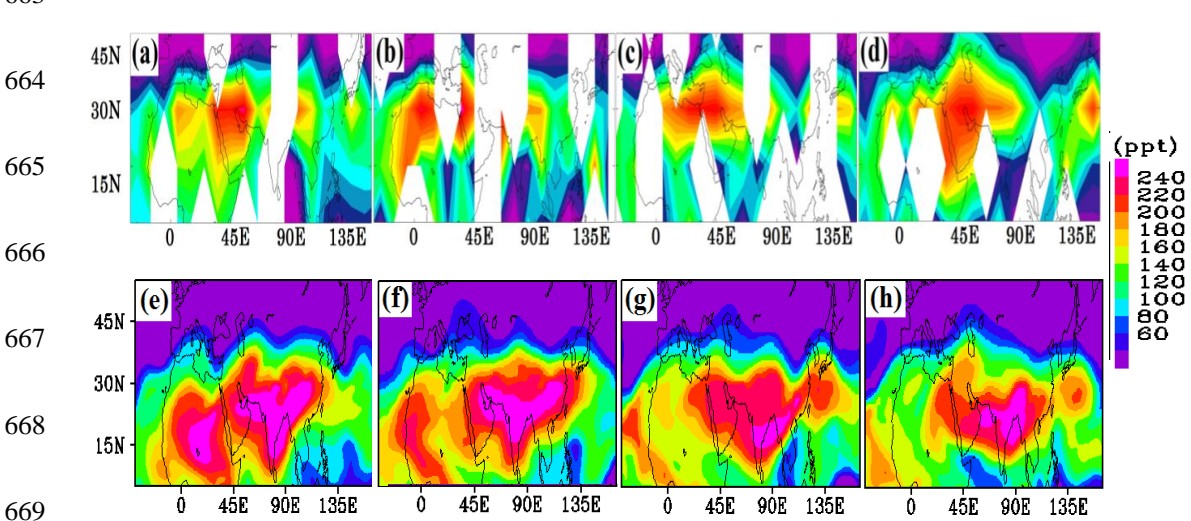

669

Figure 6: Spatial distribution of PAN mixing ratios (ppt) at 16 km: MIPAS satellite observations for (a) 1-2 July, (b) 3-4 July, (c) 5-6 July, (d) 7-8 July, 2003, and ECHAM5-HAMMOZ CTRL simulations for (e) 02 July, (f) 04 July, (g) 06 July, (h) 08 July, 2003.

673

674

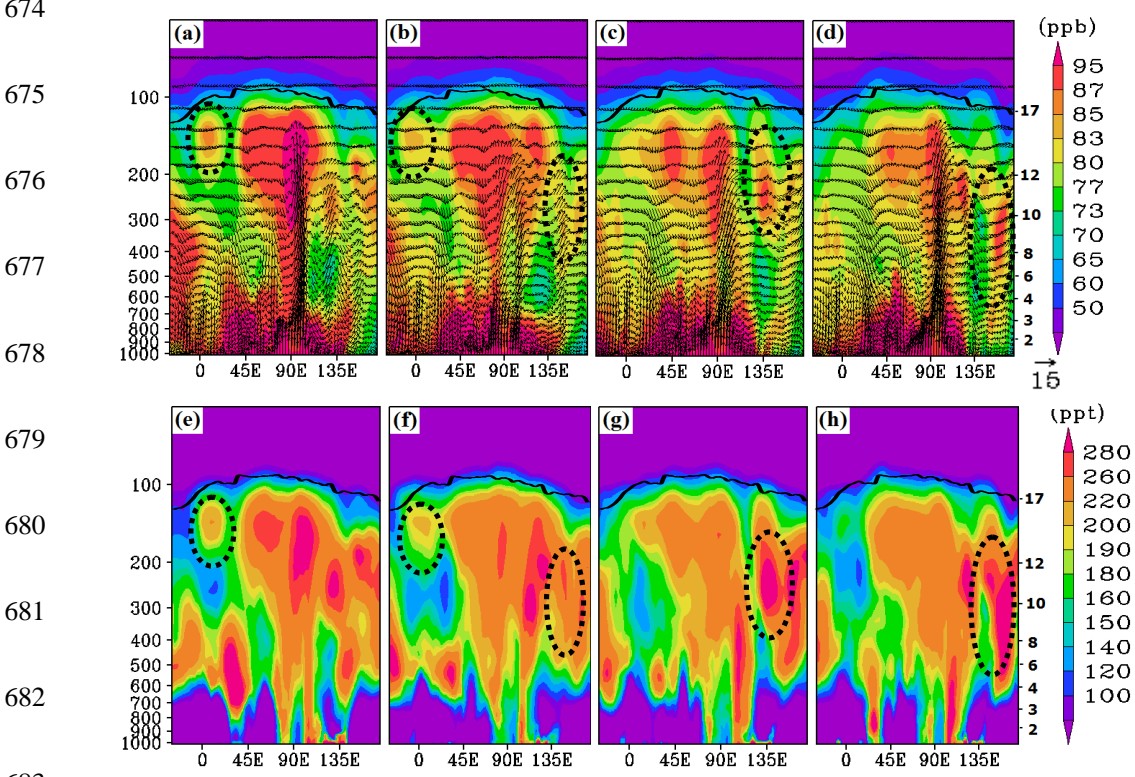


Figure 7:     Longitude-pressure section (averaged for 20°-40° N) of CO (ppb) from
ECHAM5-HAMMOZ CTRL simulation for (a) 02 July, (b) 04 July, (c) 06 July, (d) 08 July,
2003. Wind vectors (m·s$^{-1}$) are shown by black arrows. Vertical velocity field is scaled by a
factor of 300. (e)-(h) same as (a)-(d) but for PAN (ppt). Black thick line indicates tropopause and
black dotted circles indicate eddies. Pressure (hPa) is indicated on left y-axis and altitudes (km)
on right y-axis.



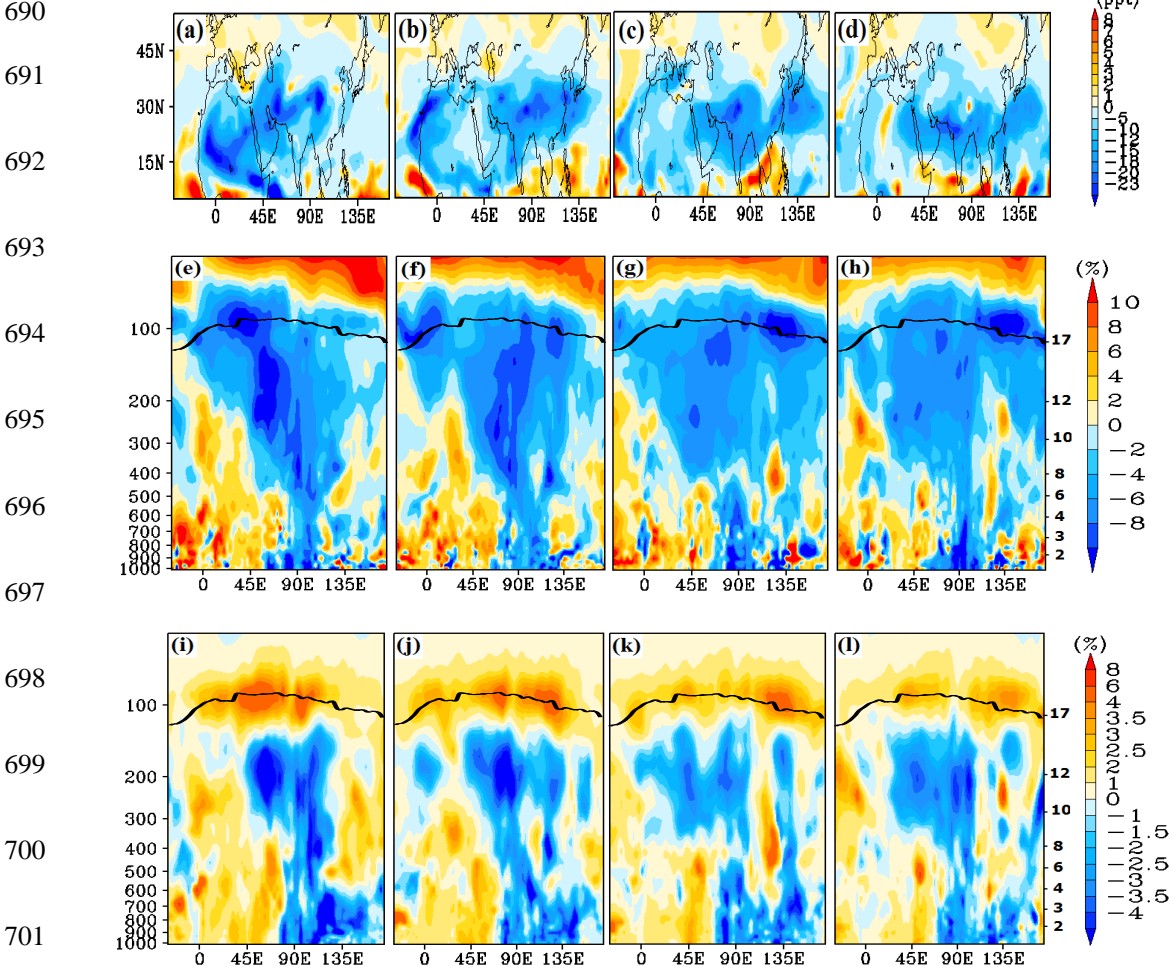

Figure 8: Spatial distribution of anomalies of PAN mixing ratios (ppt) (color shades) at 16 km

from ECHAM5-HAMMOZ model simulations for (a) 02 July, (b) 04 July, (c) 06 July, (d) 08

July, 2003. Longitude-pressure distribution (averaged for 20°-40° N) of anomalies of PAN

(%) for (e) 02 July, (f) 04 July, (g) 06 July, (h) 08 July, 2003. (i)-(l) same as (e)-(h) but for

ozone anomalies (%). Black thick line indicates tropopause. Pressure (hPa) is indicated on left

y-axis and altitudes (km) on right y-axis.





708

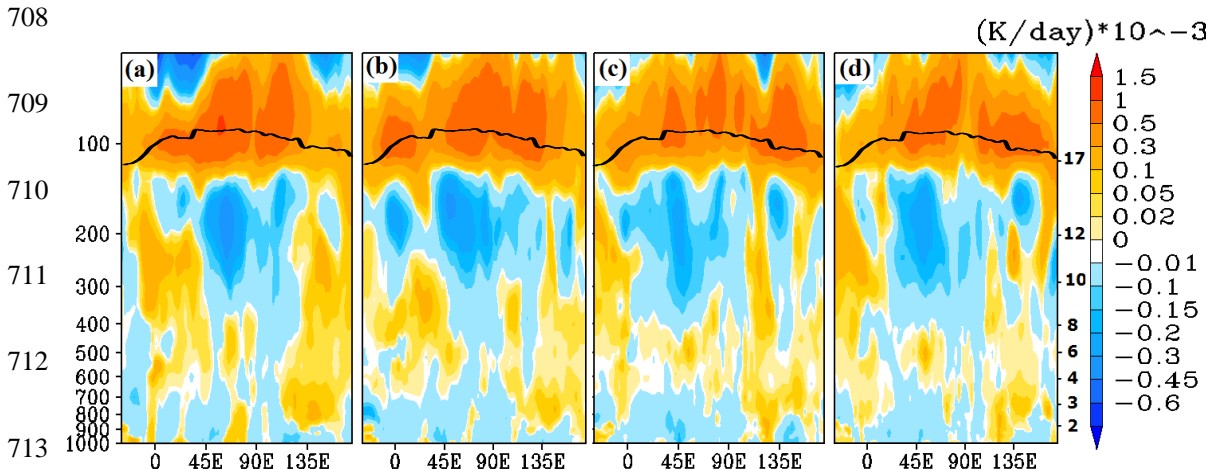

Figure 9: Longitude-pressure distribution of anomalies of ozone heating rates ((K·day$^{-1}$) ×10$^{-3}$)
for (a) 02 July, (b) 04 July, (c) 06 July, (d) 08 July, 2003. Pressure (hPa) is indicated on left y-
axis and altitudes (km) on right y-axis.