# Peer review of "Transport of Asian trace gases via eddy shedding from the Asian summer monsoon anticyclone and associated impacts on ozone heating rates"

_Atmospheric Chemistry and Physics, 2018_

## Short Comment (SC1) · 21 Mar 2018

**Comment on the manuscript "Transport of Asian trace gases via eddy shedding from the Asian summer monsoon" by S. Fadnavis et al.**

by Gabriele P. Stiller (gabriele.stiller@kit.edu)

Dear Dr. Fadnavis,

I have read with interest you paper on the transport of trace gases related to the Asian monsoon. I have noticed that you have used MIPAS data generated by my team at Karlsruhe Institute of Technology, IMK, for demonstrating observational evidence, and I am pleased about this. However, if you allow me, I would like to comment on the representation of MIPAS data in your Figures 4 to 6. I feel this representation is misleading. The interpolation routine used has a number of drawbacks:

- In my opinion, the number of data points within a geographical bin is too low that an interpolation provides a meaningful result. We have checked the MIPAS data and we find, in the relevant area, only 1 or 2 data points per 15 deg latitude x 10 deg longitude bin for most cases, and never more than 5.

- The MIPAS data are not recorded synoptically. In particular, for a monsoon pattern moving (and changing its shape) as fast as you show, it makes a huge difference if the data are recorded at the beginning of the first day or the end of the second day. This is another argument against averaging and interpolating the data.

- We have checked the positions of the individual data points against your averaged and interpolated data fields. The interpolation artificially "generates" or "invents" data at locations where there are none in the original data. One prominent example is ozone for the 1-2 July 2003: due to cloud coverage, there is a huge data gap between 20-30 N and 45-110 E; in your representation, this data gap is partly filled (see Fig. 1).

[Figure]

Figure 1: Positions of MIPAS observations and retrieved volume mixing ratios of ozone at 16 km (color scale) for 1 and 2 July 2003, for the latitude/longitude range shown in Figure 4a of Fadnavis et al., 2018.

We would like to suggest a different representation of the MIPAS data: we would find it appropriate to plot the single data points of MIPAS observations above the model fields, in the same color scale. This would provide a quantitative impression whether the model data reproduce the observations in an adequate way. We are, however, aware that the non-synoptic representation contained in the MIPAS data still cannot be avoided in this way. To overcome this problem, the model fields would need to be evaluated at the locations and times of the MIPAS observations.

In Fig. 2 we provide an example of interpolation of the MIPAS data that, in our opinion, is closer to the original data, although it also does not fully avoid to "invent" data where there are none in the original observations. This interpolation of MIPAS data has been done by calculating interpolated data on a fine (similar to model output) geographical grid. For each grid point, the surrounding MIPAS data points are averaged while applying a distance weighting. The maximum distance for which MIPAS data points are considered is ± 7 deg in latitude and ± 15 deg in longitude (covering a box of 14 deg in latitude and 30 deg in longitude), and a minimum number of 2 data points per interpolation grid point is requested. In our opinion, the result resembles the MIPAS data field much better, while, again, the problem of non-synoptical observations cannot be avoided.

Besides the comment above, I would like to mention that the referencing of MIPAS data could be improved. The reference provided (von Clarmann et al., 2009) is certainly not the one that was intended to be cited. Please find the correct reference below. The retrievals of CO and PAN are not covered by this paper. I have added the references to be cited for these two species below.

We would appreciate if you considered our comments in the revised version of your manuscript.

Sincerely,
Gabriele Stiller
for the MIPAS teams at KIT, IMK, Karlsruhe, and CSIC/IAA, Granada

**References**
von Clarmann, T., Höpfner, M., Kellmann, S., Linden, A., Chauhan, S., Funke, B., Grabowski, U., Glatthor, N., Kiefer, M., Schieferdecker, T., Stiller, G. P., and Versick, S.: Retrieval of temperature, $H_2O$, $O_3$, $HNO_3$, $CH_4$, $N_2O$, $ClONO_2$ and ClO from MIPAS reduced resolution nominal mode limb emission measurements, Atmos. Meas. Tech., 2, 159-175, doi: 10.5194/amt-2-159-2009, 2009.
Glatthor, N., von Clarmann, T., Fischer, H., Funke, B., Grabowski, U., Höpfner, M., Kellmann, S., Kiefer, M., Linden, A., Milz, M., Steck, T., and Stiller, G. P.: Global peroxyacetyl nitrate (PAN) retrieval in the upper troposphere from limb emission spectra of the Michelson Interferometer for Passive Atmospheric Sounding (MIPAS), Atmos. Chem. Phys., 7, 2775-2787, doi: 10.5194/acp-7-2775-2007, 2007.
Funke, B., López-Puertas, M., García-Comas, M., Stiller, G. P., von Clarmann, T., Höpfner, M., Glatthor, N., Grabowski, U., Kellmann, S., and Linden, A.: Carbon monoxide distributions from the upper troposphere to the mesosphere inferred from 4.7 $\mu$m non-local thermal equilibrium emissions measured by MIPAS on Envisat, Atmos. Chem. Phys., 9, 2387-2411, doi: 10.5194/acp-9-2387-2009, 2009.

[Figure]

Figure 2: MIPAS observations (in terms of vmr) of CO (left) and ozone (right) at 16 km altitude for the following days: 1 and 2 July 2003 (top row), 3 and 4 July 2003 (second row), 5 and 6 July (third row), and 7 and 8 July (bottom row). The diamonds indicate the positions of the observations and the individually retrieved values (color scale on the right of each panel). The color shading provides an interpolated field using the observed data. For description of the interpolation method, see text. The latitude/longitude range is similar to that of Figs. 4 and 5 in Fadnavis et al., 2018.

---

## Referee Comment (RC1) · Anonymous Referee #1 · 27 Mar 2018

The paper by Fadnavis et al. investigates transport of trace gases via eddy shading from the Asian summer monsoon anticyclone and associated impacts on ozone heating rates using model simulations as well as observations. The paper is generally well written and structured. However, in some parts of the paper additional information on the method applied (e. g. the power spectrum analyses) or clear discussion on some specific results (the Asia10 simulation) are missing. I would recommend some major revisions before the paper can be accepted for publication in ACP. See my detailed comments listed below.

[Figure]

**Specific comments:**

Title: "Asian trace gases"? I guess "Asian" is obsolete. It should rather read only "trace gases" or do you mean "Asian emissions"?

P2, L22: The sentence should be rephrased, "are instrumental in distributing the Asian trace gases....." sounds weird. Also here, shouldn't it rather read "Asian emissions" instead of "Asian trace gases"? I would suggest to rephrase the sentence as follows: "Our analyses indicates that eddies detached from the anticyclone serve (or are helpful) in transporting Asian emissions (or trace gases) away from the Asian region to the West Pacific........".

P2, L24: It is not clear which frequency exactly is meant. Do you mean frequency in occurrence of eddy shedding events?

P2, L26: I would suggest to rephrase the sentence as follows: "Model sensitivity experiments considering a 10% reduction.........".

P5, L96: I would suggest to rephrase the sentence as follows: "In this study, we discuss/investigate/answer the following questions: (1) how frequently did eddy shedding events occur during the last two decades.......".

P6, L111ff: That in MIPAS-E the E stands for Envisat is not explained. It is not even mentioned that MIPAS was on board of Envisat. So either this information should be added or the E should be skipped.

P6, L119: The sentence should rather read: "Here, we analyze the MIPAS observations of CO, PAN, and $O_3$ obtained during 1-8 July 2003.

P7, L129, Section header: In case of model simulations I would not call it "Experimental". Therefore, I would suggest to rename the section into "Model set-up" or "Simulation experiment set-up".

P7, L139: Here I would replace "at" with "with a" so that it reads "The model simulations were performed with a T42 spectral resolution. . . . ...".

P8, L163: Better than what? I guess you mean "best portrayed at. . ..".

Fig 1 and 2: These figures show the same fields, namely PV and winds, but at two different potential temperature levels. These two levels are just 20 K apart, but the distributions looks completely different and shows different dynamical features. I don't understand why? I would expect that the distributions at 350 and 370 K would look quite similar.

Fig 1 and 2 caption: In Fig 1 caption it is written "at the 370 K potential temperature surface, while in Fig 2 it is written "at the 370 K level". It should be done the same way for both figure captions.

Fig 2: There are no black/red/blue arrows in this figure.

P10, L200ff: Some more information on the power spectrum analysis should be given. How is it done? References? Why is it done? What kind of information does one gain from using this kind of analysis?

P10, L208: What is the purpose of a lag-lead correlation?

Figure 3: Is "cc" in Fig c and d standing for correlation coefficient? What do the three dashed lines in Fig 3 a and b show?

Figure 4: Why are there so many gaps in the MIPAS data? Why looks MIPAS so different compared to ERA-Interim and ECHAM5-HAMMOZ? Generally, Fig a-d look quite weird. Is the binning, gridding and interpolation of the data correct? Is the color bar for MIPAS exactly the same as the ones for ERA-interim and ECHAM5-HAMMOZ? It looks a bit like something went wrong. There should be enough data to get a smooth ozone distribution.

Figure 5 and 6: Same as for Figure 3, I don't understand why the figures look so weird and why there are so many data gaps.

P13, L257-258: It is correct that MIPAS has a lower spatial coverage than ERA-Interim and ECHAM5-HAMMOZ, but the temporal coverage is much better and thus covers up for the lack in spatial resolution.

P13, L269, Section header: Ozone is not shown here. So either remove ozone from the section header or add in L260 also ozone with the remark that it is shown in the supplement.

Figure 7: Why are there no wind vectors added in the PAN figures (Fig. 7 e-f)? In the CO figures the dashed lines are not that clearly visible. Here it would be better to use white instead.

P14, L274ff: Why does the ozone distribution show completely different features than PAN and CO?

P15, L296: Fig S1 shows CDNC and ICNC. How does this relate to the emission discussed here? May it be that there is a figure missing in the supplement?

P15, L309: Which model results are shown in Figure 8? I thought it were the

CTRL simulation results. Where are then the results for Asia10 simulation shown? The entire discussion on the Asia10 simulation is confusing and should be improved.

P17, L351: See my comments above on the abstract: Do you mean an occurrence frequency?

P17, L353: What exactly has been correlated with PV?

P19, L385: Why 10%? Why has this factor been chosen? This is nowhere in the paper motivated.

P19, L395-396: Small differences? Generally, I would say that the distributions of MIPAS looks quite different to ERA-Interim and ECHAM5-HAMMOZ.

**Technical corrections:**
P9, L184 and 186: space between number and unit is missing.

P12, L236: shows → show

P12, L246: The space between the number and unit is missing.

P13, L256: special → spatial

P15, L308: and the other trace gases → and other trace gases

P16, L326: showing → shows

P16, L327: indicates → indicating

P16, L331: leads → lead

Figure 8 caption: indicates tropopause → indicates the tropopause

P18, L365: during last two → during the last two

P18, L366: over last two decades → over the last two decades

P20, L401: with gratitude → with gratitude of

---

## Referee Comment (RC2) · Anonymous Referee #3 · 10 Apr 2018

Review ACPD paper (Fadnavis et al. 2018): "Transport of Asian trace gases via eddy shedding from the Asian summer monsoon anticyclone and associated impacts on ozone heating rates"

Fadnavis et al. studies the eddy shedding aerosol by ASM in UTLS. The study shows eddy shedding from the monsoon is more frequent over west-Africa vs. West-Pacific. The lag is about 3-6 days from the center of ASM to Africa and Pacific. I found this study is interesting, however I suggest major revisions before publication. Especially the causes of the ozone anomaly near the tropopause due to emission change is unclear, and this is important to understand the ozone heating rates, which is the major

conclusions of the study.

Regarding Section 4.3:

The authors did sensitivity experiment by reduce surface emissions of NOx and NMVOCs. Figure 8 and text (Section 4.3). The anomalies of chemical tracers in UTLS region are very interesting. PAN shows an extended negative anomaly in LS, which indicates cross-tropopause transport. Can you show figure 8 (e-h) for all longitudes?

I am still confused with the high anomaly of O3 near the tropopause shown inn Figure 8 (i-l). The discussions in the paper (Line 306-320) are rather vague. What causes the ozone positive anomaly in LS?

Put model data significance (e.g. dots) on top of the plots (e.g. Figure 8).

If the LS anomaly is real and significant, I guess you should be able to see better from inert tracers e.g. CO. Please add CO plots in Figure 8 as well.

Your conclusions/findings on ozone heating rate (Section 4.4) requires your under-standing and clarification of the ozone anomaly.

Regarding MIPAS:

Following the other reviewer, pls correct/improve MIPAS data display.

Regarding Section 3.2:

Please explain details of power spectrum

Regarding Model:

You define center of ASM as 85-90E, any reason you pick the 5-deg longitude within the ASM (80-120E)?

---

## Author Comment (AC1) · 31 May 2018

The paper by Fadnavis et al. investigates transport of trace gases via eddy shading from the Asian summer monsoon anticyclone and associated impacts on ozone heating rates using model simulations as well as observations. The paper is generally well written and structured. However, in some parts of the paper additional information on the method applied (e. g. the power spectrum analyses) or clear discussion on some specific results (the Asia10 simulation) are missing. I would recommend some major revisions before the paper can be accepted for publication in ACP. See my detailed comments listed below.

[Figure]

Reply: We thank the reviewer for careful reading and valuable suggestions. Suggestions given by the reviewer have been included in the revised version of the manuscript. Changes are indicated in blue color and associated line numbers are mentioned below. A copy of the revised manuscript is provided as a supplement.

(1) Specific comments: Title: "Asian trace gases"? I guess "Asian" is obsolete. It should rather read only "trace gases" or do you mean "Asian emissions"?

Reply (1): Above mentioned suggestion is incorporated in the revised manuscript. The title of the manuscript is now changed to "Transport of trace gases via eddy shedding from the Asian summer monsoon anticyclone and associated impacts on ozone heating rates". (Line 1)

(2) P2, L22: The sentence should be rephrased, "are instrumental in distributing the Asian trace gases. . ..." sounds weird. Also here, shouldn't it rather read "Asian emissions" instead of "Asian trace gases"? I would suggest to rephrase the sentence as follows: "Our analyses indicate that eddies detached from the anticyclone serve (or are helpful) in transporting Asian emissions (or trace gases) away from the Asian region to the West Pacific. .... ...".

Reply (2): Above mentioned suggestion is incorporated in the revised manuscript. It is changed as "Our analysis indicates that eddies detached from the anticyclone contribute to the transport of Asian trace gases away from the Asian region to the West-Pacific ($20°$-$30°$ N; $120°$-$150°$ E) and West-Africa ($20°$-$30°$N, $0°$-$30°$E) (Line 22).

(3) P2, L24: It is not clear which frequency exactly is meant. Do you mean frequency of occurrence of eddy shedding events?

Reply(3): The reviewer is correct. We now clarify this as "Over the last two decades, the estimated frequency in occurrence of eddy shedding events is ∼68 % towards West-Africa and ∼25 % towards the West-Pacific. (Line 24-25)

(4) P2, L26: I would suggest to rephrase the sentence as follows: "Model sensitivity

[Figure]

experiments considering a 10% reduction. . .. . .. . ..".

Reply (4): Above mentioned suggestion is incorporated in the revised manuscript. (Line 26)

(5) P5, L96: I would suggest to rephrase the sentence as follows: "In this study, we discuss/investigate/answer the following questions: (1) how frequently did eddy shedding events occur during the last two decades. . .. . ..".

Reply (5): Above mentioned suggestion is incorporated in the revised manuscript.( Line 96).

(6) P6, L111ff: That in MIPAS-E the E stands for Envisat is not explained. It is not even mentioned that MIPAS was on board of Envisat. So either this information should be added or the E should be skipped.

Reply(6): We now clarify this as "The Michelson Interferometer for Passive Atmospheric Sounding (MIPAS) on-board the European ENVIronmental SATellite (ENVISAT) (MIPAS-E). (Lines 111-112).

(7) P6, L119: The sentence should rather read: "Here, we analyze the MIPAS observations of CO, PAN, and O3 obtained during 1-8 July 2003.

(8) P7, L129, Section header: In case of model simulations I would not call it "Experimental". Therefore, I would suggest renaming the section into "Model set-up" or "Simulation experiment set-up".

Reply (7-8): Above mentioned suggestions are incorporated in the revised manuscript (Lines 118-119 and Line 132).

(9) P7, L139: Here I would replace "at" with "with a" so that it reads "The model simulations were performed with a T42 spectral resolution. .... ...".

Reply(9): Above mentioned suggestion is incorporated in the revised manuscript (Line 142).

(10) P8, L163: Better than what? I guess you mean "best portrayed at. . .." Fig 1 and 2: These figures show the same fields, namely PV and winds, but at two different potential temperature levels. These two levels are just 20 K apart, but the distributions look completely different and show different dynamical features. I don't understand why? I would expect that the distributions at 350 and 370 K would look quite similar.

Reply (10): As suggested, it is changed to "best portrayed ..." (L172). Distribution of PV looks different at 350 K potential temperature level than 370 K since the 350 K level is ~220-250 hPa while 370 K lies near the tropopause (Fadnavis and Chattopadhyay, 2017). Distribution of PV near 370 K shows the dynamic variability of anticyclone (Fig 1) while 350 K indicates Rossby Wave Breaking (RWB) which occurs in the subtropical jet (core ~200-220 hPa). RWB is also evident in 200 hPa wind anomalies (Fig.2). It is already mentioned in the manuscript (see discussions at Lines172-175 and 192-194).

(11) Fig 1 and 2 caption: In Fig 1 caption it is written "at the 370 K potential temperature surface, while in Fig 2 it is written, "at the 370 K level". It should be done the same way for both figure captions. Fig 2: There are no black/red/blue arrows in this figure.

Reply (11): Above mentioned suggestion is incorporated in the revised manuscript. (Lines 700-701).

(12) P10, L200ff: Some more information on the power spectrum analysis should be given. How is it done? References? Why is it done? What kind of information does one gain from using this kind of analysis?

Reply (12): The power spectrum analysis (PSA) performs the temporal-to-frequency transformation via the Fast Fourier Transform (FFT). PSA gives a representation of the magnitude of the various frequency components of a signal. By looking at the spectrum, one can find how much energy or power (square of variance) is contained in the frequency components of the signal. Power spectrum simply answers the question "How much power is contained in the frequency components of the signal?" By performing PSA, some important features of signals can be discovered that are not obvious in the time series of the signal (Pelletier, 1998). For example, the local maxima in the power-spectrum (usually represented by the ordinate) would easily show what the dominant frequency component in the signal is. The dominant frequency component above the noise (the colored lines in Fig.3a,b) allow one to infer the dominant periodicity in the signal. In the context of the Fig.3a, it can be said that PV over the west Pacific contains dominant periodicity 3-5, 12-15, and ∼18-21 days.

We agree that further clarifications are needed in the text and we now add the following at lines 211-217:

"A power spectrum analysis (PSA) has been performed on the PV data (averaged for 300-100 hPa) during 1995-2016 for West-Africa (20-30°N, 0-30°E) and the West-Pacific (20-30°N, 140-150°E). The PSA uses the temporal-to-frequency fast Fourier transform in order to identify dominant signal frequencies. It provides information of signal power (square of variance) associated with the frequency components of the signal, with the dominant signal periodicity being the inverse of the dominant signal frequency. Figures 3a-b show the distribution of power spectral variance over the West-Africa and West-Pacific regions. The variance corresponding to the periodicities 3-5 days and 12-15, 18-21 days is significant at 95 % confidence level for both the regions.

(13) P10, L208: What is the purpose of a lag-lead correlation?

Reply(13): The usefulness of the lagged scatter plot is that it can display peak cross-correlation (positive or negative) at a certain lag k, where lag k denotes the time scale of response or cause-effect relationship.

We compute lag-lead correlation between two time series (1995-2015) of PV.

Case-I: Time series over (1) West-Africa (20-30N, 0-30E) (TSWA) and time series over (2) center of the anticyclone (85-90E, 28-30N) (TSC).

Case-II: Time series over (1) West-Pacific (20-30N, 140-150E) (TSWP) and time series over (2) center of the anticyclone (85-90E, 28-30N).

Fig. 3c indicates that TSWA show lead-correlation with TSC (peak) by 3-5 days, imply-ing eddies over TSWA could originate from ASM anticyclone reaching west-Africa after 3-5 days as seen in Figure 1.

Similarly, for case-II, Fig. 3d indicates that time series of PV over TSWP show lead-correlation with TSC by 5-6 days, implying eddies over TSWP could originate from ASM anticyclone reaching over the west Pacific after 5- 6 days as seen in Figure 1.

It is discussed at L222-230 as "The lag-lead correlation of PV (averaged for 200-100 hPa) for the center region of the anticyclone (85-90° E, 28-30° N) with PV averaged over the West-Pacific shows a maximum positive lead correlation at 3-4 days (Fig. 3c). Similarly, PV over West-Africa shows a maximum positive lead correlation for 5-6 days with the PV averaged over the monsoon anticyclone (Fig. 3d). This indicates that the transport of the eddies from the anticyclone (source region) has a typical duration of three to four days over the West Pacific and five to six days over West Africa. This transport time is the timescale over which the trace gases are moved to remote loca-tions from the ASM anticyclone".

(14) Figure 3: Is "cc" in Fig c and d standing for correlation coefficient? What do the three dashed lines in Fig 3 a and b show?

Reply(14): Thank you for the suggestion. Yes, "cc" stands for standard Pearson sample linear correlation coefficient. The three lines are the theoretical Markov spectrum (mid-dle, green) and the lower (blue curve 5%) and upper (red curve 95%) confidence level using the lag-1 autocorrelation. Any spectral peaks above the red curve are statistically significant. We now add these suggestions in Figure 3 and caption. (Lines720-722).

(15) Figure 4: Why are there so many gaps in the MIPAS data? Why looks MIPAS so different compared to ERA-Interim and ECHAM5-HAMMOZ? Generally, Fig a-d look quite weird. Is the binning, gridding and interpolation of the data correct? Is the color bar for MIPAS exactly the same as the ones for ERA-interim and ECHAM5- HAMMOZ? It looks a bit like something went wrong. There should be enough data to get a smooth

ozone distribution.

(16) Figure 5 and 6: Same as for Figure 3, I don't understand why the figures look so weird and why there are so many data gaps.

(17) P13, L257-258: It is correct that MIPAS has a lower spatial coverage than ERA-Interim and ECHAM5-HAMMOZ, but the temporal coverage is much better and thus covers up for the lack of spatial resolution.

Reply(15-17): Above mentioned sentence is removed. Figures 5 and 6 are now re-plotted with data provided by MIPAS-Team. Some gaps are inevitable as MIPAS cannot measure in the presence of clouds. (Pages 32-34).

(18) P13, L269, Section header: Ozone is not shown here. So either remove ozone from the section header or add in L260 also ozone with the remark that it is shown in the supplement.

Reply (18): In this section, we have discussed the distribution of ozone at Lines 296-304. We have added Fig S2 and related discussion as "The vertical distribution of ozone shows low ozone amounts extending from the convective regions of Bay of Bengal (80-95° E) and South China Sea ($\sim$120° E) upward in the upper troposphere (Figs. S2a-h), with ozone amounts of $\sim$100-150 ppb near the tropopause (see also Figs. 4-i-l). The lower ozone amounts over the Asian troposphere may be due to clear marine air masses during the monsoon season (Zhao et al., 2009). The feature of low ozone air-mass ascent is less evident than the CO and PAN vertical ascent, due to a number of factors which are influencing ozone production and loss processes at different altitudes in the troposphere and lower stratosphere, such as stratospheric intrusions, lightning etc. (see discussions in section 3.4)". Therefore ozone is retained in the section header.

(19) Figure 7: Why are there no wind vectors added in the PAN figures (Fig. 7 e-f)? In the CO figures the dashed lines are not that clearly visible. Here it would be better to

use white instead.

Reply(19): Thank you for the suggestion. As suggested wind vectors are plotted in Fig. 7 e-f. We have tried to change the color of the dashed line to white. However, it is still not clear. To improve the clarity we have replaced dotted lines with continuous black lines. (page 35).

(20) P14, L274ff: Why does the ozone distribution show completely different features than PAN and CO?

Reply(20): First, ozone is a stratospheric tracer gas, while PAN and CO are tropospheric gases. Therefore differences are expected. Second, we have now incorporated supplementary figures and discussions to justify the variation of ozone anomalies near the tropopause (revised Fig-l, Fig. S2, and Fig S3 and discussions at Lines 334-352).

Figures S3e-h show the distribution of ozone anomalies in the monsoon anticyclone (∼100 hPa). The spatial distribution indicates that the response to emission reductions generates negative anomalies of ozone in the southern part of anticyclone (15-25°N; 60-120°E), while ozone anomalies are positive in the northern part of the anticyclone. The positive ozone anomalies in the northern part of the anticyclone may be a result of a weaker transport of Asian emissions there (Fig. 4). Figs. 8i-l and S3a-d also show positive anomalies of ozone near the tropopause, which are also likely to be a response to changes in dynamics due to emission changes, e.g., stratospheric intrusions (∼100°E, 30-40°N as illustrated by Figures S2a-h and S3a-d) along the subtropical jet at the northern flank of the anticyclone enhancing ozone (Fadnavis and Chattopadhyay, 2017). The ozone variability near the tropopause is generally driven by the strong mixing of tropospheric and stratospheric air-masses. However, such analysis is beyond the scope of the paper.

(21) P15, L296: Fig S1 shows CDNC and ICNC. How does this relate to the emission discussed here? May it be that there is a figure missing in the supplement?

Reply(21): Here we discuss transport of emission due to convection. To show regions of convection we have plotted cloud droplet (CDNC) and ice crystal (ICNC) number concentrations (in mg-1) in Fig. S1 (Lines 284-286).

This sentence is re-written as "The location of the plume (Fig. 7) coincides with a strong convection region - see Fig. S1, showing combined cloud droplet (CDNC) and ice crystal (ICNC) number concentrations from the CTRL simulation. This indicates that surface emissions are lifted up by the monsoon convection.

(22) P15, L309: Which model results are shown in Figure 8? I thought it was the CTRL simulation results. Where are then the results of Asia10 simulation shown? The entire discussion on the Asia10 simulation is confusing and should be improved.

Reply(22): Figure 8 shows anomalies in ozone obtained from (Asia10 – CTRL) depicting the impact of emission perturbation for Asian NMVOCs and NOx. Discussion in this regard is now incorporated at Lines 150-152 for further clarifications.

(23) P17, L351: See my comments above on the abstract: Do you mean an occurrence frequency?

Reply (23): Yes, we do. The above-mentioned suggestion is incorporated in the revised manuscript (Line 401).

(24) P17, L353: What exactly has been correlated with PV?

Reply(24): This sentence is re-written. In the UTLS (300-100 hPa), eddies (PV<2 PUV) over West-Africa (3-4 days) and West-Pacific (5-6 days) show lead correlation with the center of the anticyclone. It indicates that the anticyclone sheds eddies with transport duration of typically three to four days to West Africa and five-six days to the Western Pacific (Lines 403-405).

(25) P19, L385: Why 10%? Why has this factor been chosen? This is nowhere in the paper motivated.

[Figure]

Reply (25): A rate of increase of every species of NMVOCs varies with time and regions in Asia (Li et al 2014). This fixed 10% reduction was chosen due to the spatial-temporal variability of NMVOCs over Asia and the inherent difficulty in obtaining a common trend value (Li et al 2014).. These simulations are adopted from Fadnavis et al., (2015). Flat 10% reduction of Asian NOx emissions in sensitivity experiments were also reported by Naik et al., (2005).

Naik, V., D. Mauzerall, L. Horowitz, M. D. Schwarzkopf, V. Ramaswamy, and M. Oppenheimer (2005), Net radiative forcing due to changes in regional emissions of tropospheric ozone precursors, J. Geophys. Res., 110, D24306, doi:10.1029/2005JD005908.

In the revised version (Lines 147-152), we have given a motivation for Asia10 simulation.

(26) P19, L395-396: Small differences? Generally, I would say that the distributions of MIPAS look quite different to ERA-Interim and ECHAM5-HAMMOZ.

Reply(26): As suggested, an above-mentioned sentence is modified. (Line 269).

Technical corrections:

1. P9, L184 and 186: space between number and unit is missing. (see Lines 193 and 195)

2. P12, L236: shows → show (This sentence is reframed see Lines 252–253)

3. P12, L246: The space between the number and unit is missing. P13, L256: special → spatial (see Lines 262 and 272)

4. P15, L308: and the other trace gases → and other trace gases (see Lines363)

5. P16, L326: showing → shows (see Line 373)

6. P16, L327: indicates → indicating (see Line 374)

7. P16, L331: leads → lead Figure 8 caption: indicates tropopause → indicates the tropopause P18, L365: during last two → during the last two (see Lines 378 and 798 and 415)

8. P18, L366: over last two decades → over the last two decades (this sentence is rephrased see Line 416)

9. P20, L401: with gratitude → with gratitude of (see Line 452)

Reply(1-9): all the above technical corrections are included in the revised version of the manuscript.

Please also note the supplement to this comment:
https://www.atmos-chem-phys-discuss.net/acp-2018-168/acp-2018-168-AC1-supplement.pdf

⎯⎯⎯⎯⎯⎯⎯⎯⎯⎯⎯⎯

---

## Author Comment (AC2) · 31 May 2018

"Transport of Asian trace gases via eddy shedding from the Asian summer monsoon anticyclone and associated impacts on ozone heating rates" Fadnavis et al. studies the eddy shedding aerosol by ASM in UTLS. The study shows eddy shedding from the monsoon is more frequent over west-Africa vs. West-Pacific. The lag is about 3-6 days from the center of ASM to Africa and Pacific. I found this study is interesting; however, I suggest major revisions before publication. Especially the causes of the ozone anomaly near the tropopause due to emission change is unclear, and this is important to understand the ozone heating rates, which is the major conclusions of the

study.

Reply: We thank the reviewer for the positive comments and valuable suggestions. We have now added supplementary figures and discussions to explain the variation of ozone anomalies near the tropopause. Changes are indicated in blue color and associated line numbers are mentioned below. A copy of the revised manuscript is provided as a supplement (also Fig.R1 and Fig. R2)

(1) Regarding Section 4.3: The authors did sensitivity experiment by reducing surface emissions of NOx and NMVOCs. Figure 8 and text (Section 4.3). The anomalies of chemical tracers in UTLS region are very interesting. PAN shows an extended negative anomaly in LS, which indicates cross-tropopause transport. Can you show figure 8 (e-h) for all longitudes? I am still confused with the high anomaly of O3 near the tropopause is shown in Figure 8 (i-l). The discussions in the paper (Line 306-320) are rather vague. What causes the ozone positive anomaly in LS?

Reply(1):We have now incorporated supplementary figures and discussions to elaborate on the variation of ozone anomalies near the tropopause (revised Fig-8i-l, Fig.S2-Fig.S4 and discussions at Lines 334-360).

We have plotted Figure 8(e-h) for all the longitudes (shown as Fig. R1). This indicates that the negative anomalies of PAN extend to ∼20W, which is why we decided to choose this value as the western most longitude in our figures.

(2) Put model data significance (e.g. dots) on top of the plots (e.g. Figure 8). If the LS anomaly is real and significant, I guess you should be able to see better from inert tracers e.g. CO. Please add CO plots in Figure 8 as well.

Reply(2): Our model simulations (nudged runs) are forced with meteorology (vorticity, surface pressure, divergence, and temperature). Since simulations are driven by above mentioned meteorological fields from European Centre for Medium-Range Weather Forecasts operational analyses (Integrated Forecast System (IFS) cycle-32r2), every

six hours during 2003, model's mean state is forced towards the real atmospheric condition of 2003 (see discussions on Lines 153-155). Therefore simulations give representation atmospheric conditions of 2003 and therefore are supposed to be significant.

The significance test is applied to the free runs (not forced with meteorology) which are conducted for numbers of years (not for a particular year, unlike 2003 in the current paper). Since they are free runs, it is important to show statistical significance. The numbers of years are the numbers of samples on which one applies the significance test. In the case of current model set-up, numbers of samples are not available to apply significance test (only one sample which represents meteorological state atmosphere during 2003). Figure 8 shows 6-8% reduction in PAN and O3 in response to 10% reduction of Asian NOx and NMVOCs which is quite significant.

In emission perturbation experiments, we have changed Asian emissions of NOx and NMVOCs. Therefore variations in CO are subject to many different influences and thus not clear (figure shown as Fig. R2).

(3) Your conclusions/findings on ozone heating rate (Section 4.4) require your understanding and clarification of the ozone anomaly.

Reply(3): We have incorporated discussions (Lines 334-360).

(4) Regarding MIPAS: Following the other reviewer, pls correct/improve MIPAS data display.

Reply (4): We have now updated the figures with the data re-gridded by the MIPAS team members. Therefore data gaps are less (but still present due to presence of clouds) in the updated figures.

(5) Regarding Section 3.2: Please explain details of a power spectrum

Reply(5): Thank you for the suggestions. As suggested details of the power spectrum analysis are incorporated in the revised manuscript (Lines 211-L217). "The PSA uses the temporal-to-frequency fast Fourier transform in order to identify dominant signal

frequencies. It provides information of signal power (square of variance) associated with the frequency components of the signal, with the dominant signal periodicity being the inverse of the dominant signal frequency. Figures 3a-b show the distribution of power spectral variance over the West-Africa and West-Pacific regions. The variances corresponding to the periodicities of 3-5 days, 12-15, and 18-21 days are significant at 95 % confidence level for both the regions indicating that the eddy shedding activity is dominated in the range of synoptic frequency ($\sim$10 days).:

(6) Regarding Model: You define a center of ASM as 85-90E, any reason you pick the 5-deg longitude within the ASM (80-120E)?

Reply(6): The 85áţŠ-90áţŠE; 28áţŠ-30áţŠN is the core Tibetan anticyclone zone where the center of the climatological monsoon (Tibetan) anticyclone is located. Therefore it is taken as a representative region of Tibetan anticyclone zone. Also, the monsoon anticyclone is highly dynamic in nature with respect to its position and shape (Popovic and Plumb, 2001; Garny and Randel, 2013; Vogel et al., 2016) also see Fig.1. Therefore we have defined a center of ASM 5-deg wide, as 85-90E. (Lines 222-224)

Please also note the supplement to this comment:
https://www.atmos-chem-phys-discuss.net/acp-2018-168/acp-2018-168-AC2-supplement.zip

―――――――――――――――――

[Figure]

[Figure]

Figure R1: Longitude-pressure distribution (averaged for 20°-40° N) of anomalies of PAN (%) for (e) 02 July, (f) 04 July, (g) 06 July, (h) 08 July 2003.The black line indicates the tropopause. Pressure (hPa) is indicated on left y-axis and altitudes (km) on the right y-axis.

**Fig. 1.**

[Figure]

**Figure R2**: Latitude-pressure section of CO anomalies (averaged for 18-20°N) expressed as % change, (a) 02 July, (b) 04 July, (c) 06 July, (d) 08 July, 2003. Black thick line indicates the tropopause.

**Fig. 2.**

---

## Author Comment (AC3) · 31 May 2018

I have read with interest you paper on the transport of trace gases related to the Asian monsoon. I have noticed that you have used MIPAS data generated by my team at Karlsruhe Institute of Technology, IMK, for demonstrating observational evidence, and I am pleased about this. However, if you allow me, I would like to comment on the representation of MIPAS data in your Figures 4 to 6. I feel this representation is misleading. The interpolation routine used has a number of drawbacks:

In my opinion, the number of data points within a geographical bin is too low that an interpolation provides a meaningful result. We have checked the MIPAS data and we

find, in the relevant area, only 1 or 2 data points per 15 deg latitude x 10 deg longitude bin for most cases, and never more than 5.

The MIPAS data are not recorded synoptically. In particular, for a monsoon pattern moving (and changing its shape) as fast as you show, it makes a huge difference if the data are recorded at the beginning of the first day or the end of the second day. This is another argument against averaging and interpolating the data.

We have checked the positions of the individual data points against your averaged and interpolated data fields. The interpolation artificially "generates" or "invents" data at locations where there are none in the original data. One prominent example is ozone for the 1-2 July 2003: due to cloud coverage, there is a huge data gap between 20-30 N and 45-110 E; in your representation, this data gap is partly filled (see Fig. 1). Figure 1: Positions of MIPAS observations and retrieved volume mixing ratios of ozone at 16 km (color scale) for 1 and 2 July 2003, for the latitude/longitude range shown in Figure 4a of Fadnavis et al., 2018. 1 We would like to suggest a different representation of the MIPAS data: we would find it appropriate to plot the single data points of MIPAS observations above the model fields, in the same color scale. This would provide a quantitative impression whether the model data reproduce the observations in an adequate way. We are, however, aware that the non-synoptic representation contained in the MIPAS data still cannot be avoided in this way.

To overcome this problem, the model fields would need to be evaluated at the locations and times of the MIPAS observations. In Fig. 2 we provide an example of interpolation of the MIPAS data that, in our opinion, is closer to the original data, although it also does not fully avoid to "invent" data where there are none in the original observations. This interpolation of MIPAS data has been done by calculating interpolated data on a fine (similar to model output) geographical grid. For each grid point, the surrounding MIPAS data points are averaged while applying a distance weighting. The maximum distance for which MIPAS data points are considered is $\pm$ 7 deg in latitude and $\pm$ 15 deg in longitude (covering a box of 14 deg in latitude and 30 deg in longitude), and a minimum
number of 2 data points per interpolation grid point is requested. In our opinion, the result resembles the MIPAS data field much better, while, again, the problem of non-synoptical observations cannot be avoided. Besides the comment above, I would like to mention that the referencing of MIPAS data could be improved. The reference provided (von Clarmann et al., 2009) is certainly not the one that was intended to be cited. Please find the correct reference below. The retrievals of CO and PAN are not covered by this paper. I have added the references to be cited for these two species below. We would appreciate if you considered our comments in the revised version of your manuscript.

Reply: Authors are indebted to Dr. Bernd Funke, Dr. Michael Kiefer and Dr. Gabriele Stiller, Karlsruhe Institute of Technology, Germany, for helpful discussions on handling of MIPAS data and providing gridded MIPAS data for the current study. We have used the gridded data provided by them for creating figures 4-6 in the revised manuscript. We have duly acknowledged them for their support.

---

## Author Response (AR2)

Suggestions for revision or reasons for rejection (will be published if the paper is accepted for final publication)

We thank the reviewer for careful reading and suggestions. We have incorporated all the suggestions given by the reviewer. The changes are indicated in blue color and corresponding line numbers are also mentioned in the replies given below.

(1) P7, L139: space missing between "and" and "168" .

Reply(1) : Above mentioned suggestion has been incorporated in the manuscript at L139.

(2) P13, L252: add "the" so that it reads "the model overestimates".

Reply(2) : Above mentioned suggestion has been incorporated in the manuscript at L252.

(3) P13, L259-260. This sentence is somewhat weird. It should be rephrased. What do you mean with "a location of maximum"? Do you mean "the location of maximum in the CO (distribution)? Wouldn't it then be better to write "Similar to ozone the maximum in the CO distribution is not collocated with......". The second part of the sentence is also not clear and thus should be rather an own sentence. One could continue: "Further, slight differences between model simulations and MIPAS observations are found." Why are there differences between model simulation and observation? What is the reason for these differences?

Reply(3) : Above mentioned sentence has been reframed and the reasons for differences between model and observations is explained in the manuscript at L259-262.

(4)  P16, l326: long lived -> long-lived

Reply(4) : Above mentioned suggestion has been incorporated in the manuscript at L327.

(5) P16, L326: studeies -> studies

Reply(5) : This word has been removed from the manuscript.

(6) P16, L326: This sentence should also be rephrased. What do you mean with should behave in the model simulation similar to inert gases? I guess it should rather read "model simulation" than "model studies".

Reply(6) : Above mentioned suggestion has been incorporated in the manuscript at L328.

(7) P16, L333: indicated by boxes? Where in the figure? Please clearly state that.

Reply(7) : Above mentioned suggestion has been incorporated in the manuscript at L335.

(8) P17, L344-345: "of from" should be either "of" or "from".

Reply(8) : Above sentence has been reframed as "Ozone distributions from CTRL simulations show stratospheric intrusion" at L346-347.

(9) P17, L347: add "the" so that it reads "in the Asia10 simulations...".

Reply(9) : Above mentioned suggestion has been incorporated in the manuscript at L348.

(10) P17, L353: add "the" so that it reads "from the surface to ~180 hPa....

Reply(10) : Above mentioned suggestion has been incorporated in the manuscript at L355.

(11) P17, L354: skip "be" after may and "to" after likely, so that it reads "may likely be".

Reply(11) : Above mentioned suggestion has been incorporated in the manuscript at L356.

(12) P17, L357: It -> This

Reply(12) : Above mentioned suggestion has been incorporated in the manuscript at L359

(13) P18, L365: add "in the simulation" so that it reads "in the CTRL simulation".

Reply(13) : Above mentioned suggestion has been incorporated in the manuscript at L367.

(14) P18, L376: also see -> see also

Reply(14) : Above mentioned suggestion has been incorporated in the manuscript at L379.

(15) P19, L392: show -> showed

Reply(15) : Above mentioned suggestion has been incorporated in the manuscript at L395.

(16) P20, L404: space between 5-6 and days is missing.

 Reply(16) : Above mentioned suggestion has been incorporated in the manuscript at L407-408.

(17) P20, L407: employ -> employed

Reply(17) : Above mentioned suggestion has been incorporated in the manuscript at L410.

(18) P20, L417: evaluate -> evaluated

Reply(18) : Above mentioned suggestion has been incorporated in the manuscript at L420.

(19) P21, L431ff: this text part needs to be rephrased, too. How can emission change dynamics? What do you mean with "induced by reduced of NOx and OH? Do you mean by reduced emissions or do you mean by lower concentrations of NOx and OH in the simulation?

Reply(19) : Above mentioned suggestion has been incorporated in the manuscript at L433.

(20) P21, L440:show -> showed L443

Reply(20) : Above mentioned suggestion has been incorporated in the manuscript at L443.

(21) P22, L448: One fullstop obsolete.

Reply(21) : Above mentioned suggestion has been incorporated in the manuscript at L451.

(22) P22, L448-449: these leaves the question open why are there differences found. What is the reason for the differences?

Reply(22) : The reasons for observed differences are explained in the revised manuscript at L451-454.